# Molecular Characterizations of Gynecologic Carcinosarcomas: A Focus on the Immune Microenvironment

**DOI:** 10.3390/cancers14184465

**Published:** 2022-09-14

**Authors:** Sanaa Nakad Borrego, Ernst Lengyel, Katherine C. Kurnit

**Affiliations:** Department of Obstetrics and Gynecology, Section of Gynecologic Oncology, The University of Chicago, Chicago, IL 60637, USA

**Keywords:** gynecologic carcinosarcoma, tumor immune microenvironment, immunotherapy

## Abstract

**Simple Summary:**

Gynecologic carcinosarcomas are rare and highly aggressive tumors. Despite treatment, response rates are low, and the survival outcomes of patients with carcinosarcoma are far worse than high-grade tumors of the same origin. Due to this, there is a need for new, tailored therapies. The role of immunotherapy has not been extensively studied in gynecologic carcinosarcomas, but emerging studies suggest that there is a potential role that should be explored further. The aim of this review is to outline the current knowledge about gynecologic carcinosarcomas, with a focus on their molecular profiles and the tumor immune microenvironment, and discuss possible directions for future research.

**Abstract:**

Gynecologic carcinosarcomas, specifically of endometrial and ovarian origin, are aggressive and rare tumors. Treatment data are limited and are often extrapolated from other histologies and smaller retrospective studies. While the optimal therapy approach remains contentious, treatment is often multimodal and may include surgery, chemotherapy, radiation, or a combination of multiple strategies. However, despite aggressive treatment, these tumors fare worse than carcinomas of the same anatomic sites irrespective of their stage. Recent studies have described in-depth molecular characterizations of gynecologic carcinosarcomas. Although many molecular features mirror those seen in other uterine and ovarian epithelial tumors, the high prevalence of epithelial-mesenchymal transition is more unique. Recently, molecular descriptions have expanded to begin to characterize the tumor immune microenvironment. While the importance of the immune microenvironment has been well-established for other tumor types, it has been less systematically explored in gynecologic carcinosarcomas. Furthermore, the use of immunotherapy in patients with gynecologic carcinosarcomas has not been extensively evaluated. In this review, we summarize the available data surrounding gynecologic carcinosarcomas, with a focus on the immune microenvironment. We end with a discussion of potential immunotherapy uses and future directions for the field.

## 1. Introduction

Carcinosarcomas are biphasic tumors composed of an epithelial component and a mesenchymal component (Figure 1, images obtained under The University of Chicago IRB-approved protocol) [1]. These most commonly arise from the uterus or ovary but can arise from other organs as well, such as the lung or breast [2,3]. Gynecologic carcinosarcomas are rare but aggressive, comprising less than 5% of uterine and ovarian cancers. However, relative to other high-grade uterine cancers, survival outcomes are much worse [4]. The same is true for ovarian carcinosarcomas relative to high-grade serous ovarian cancers, even when matched for clinical stages [5].

Treatment algorithms have evolved over time to reflect our evolving understanding of tumor biology. Historically, these tumors were initially considered to be a type of sarcoma. However, recent molecular data that are reviewed in this article suggest that these tumors are more likely epithelial tumors that have undergone significant de-differentiation. The current biological theory posits that these tumors are either derived from a single progenitor that gives rise to carcinoma and sarcoma cells or from one progenitor carcinoma cell that undergoes a sarcomatous transformation [6,7]. These tumors are now considered epithelial in origin, and this designation has led to updated treatment algorithms that are similar to those of other epithelial endometrial and ovarian tumors.

The site of origin influences the clinical treatment algorithms for patients with gynecologic carcinosarcomas. For uterine carcinosarcoma, surgical resection with staging and debulking when feasible in advanced-stage patients, remains the initial treatment for the majority of patients [8]. Given the propensity for recurrence and distant spread, even in early-stage disease, the standard-of-care treatment also includes systemic chemotherapy [9]. Historically, patients with gynecologic carcinosarcomas were treated with sarcoma regimens and were enrolled in sarcoma clinical trials. Due to its historical association with other sarcomas, the previously preferred treatment of uterine carcinosarcoma was an ifosfamide doublet [10,11]. Recently, however, Gynecologic Oncology Group (GOG) 261 demonstrated that the combination of carboplatin and paclitaxel was not inferior to the combination of paclitaxel and ifosfamide [12]. This is now the first line systemic treatment of uterine carcinosarcoma similar to other endometrial carcinomas. Prior clinical trials frequently also included radiation therapy as part of adjuvant therapy. Although this has not been consistently associated with improved overall survival, radiation therapy has been associated with decreases in local recurrences and, therefore, remains a mainstay of treatment for many patients with pelvic-confined disease [1,8,13]. Unfortunately, response rates to cytotoxic chemotherapy in the second-line setting are poor, and this likely accounts for their poor survival outcomes [4,5]. However, treatment recommendations have recently begun to evolve, and cytotoxic chemotherapy is no longer the only treatment option. Immunotherapy is increasingly being used in the treatment of recurrent endometrial cancer patients as a standard-of-care option, and this practice shift has extended to some patients with uterine carcinosarcoma.

## 2. Molecular Characterization

### 2.1. Similarities to Carcinomas

The monoclonal origin of the epithelial and mesenchymal compartments has been long established in the literature [6,14]. Prior to the wide availability of whole-genome data, immunohistochemical and mutational analyses demonstrated a high concordance between both compartments [14,15]. There are reports of cases where the above similarities do not hold, which may represent true biclonal “collision tumors” [14,15]. This minority of tumors likely does evolve from two separate progenitor cells, one epithelial and one mesenchymal, that merge to evolve into a single carcinosarcoma tumor. However, this subset represents a small subset of tumors. Furthermore, the prognostic and therapeutic implications of having a monoclonal versus a biclonal tumor remain unclear [16], and thus our review primarily focuses on the instances where a single progenitor cell exists.

More recently, the next-generation sequencing and molecular profiling of carcinosarcomas have provided a better understanding of the molecular signature of gynecologic carcinosarcomas. The most frequently occurring mutations include *TP53*, *PIK3CA*, *FBXW7*, *PTEN*, *KRAS*, *CCNE1*, *PPP2R1A*, *CHD4*, and *HER2* amplification, among others [17,18]. A summary of the mutational landscape of uterine carcinosarcomas is listed in Table 1. However, the exact percentages of mutations vary across studies, likely reflecting the heterogeneous nature of these tumors [17,19,20,21,22,23,24]. For example, *TP53* is consistently the most commonly mutated gene across studies of uterine carcinosarcomas, but the rate varies from 62–91% [25,26]. In uterine carcinosarcomas, the lower end of this range (62%) was found in a study by Gotoh et al. where 35% of the uterine carcinosarcomas included had a low-grade epithelial component [17]. This is in contrast to the higher end of the range (91%), which was found in a study by Cherniack et al., where only 11% of the uterine carcinosarcomas included had a low-grade carcinoma component [20]. This mirrors the *TP53* mutation rates found in non-carcinosarcoma endometrial tumors, which are on the order of 10–20% for endometrioid endometrial cancer [27,28], in contrast with mutation rates of approximately 90% in some high-grade endometrial carcinoma histologies [29].

These findings reflect a general trend where the mutational profiles of the carcinosarcoma reflect those of tumors with the corresponding carcinoma component [6,21,30]. Zhao et al. analyzed the mutational profiles of carcinosarcomas with serous and endometrioid carcinoma components and compared them with the profiles of carcinomas with the same histology as the carcinomatous component. They identified eight driver genes with root mutations in their carcinosarcoma samples and analyzed the fraction of tumors with mutations in these genes [21]. They reported that in uterine carcinosarcomas with an endometrioid epithelial part, mutations in *PTEN*, *KRAS*, *ARID1A*, and *PIK3CA* were prevalent, whereas in uterine carcinosarcomas with a serous epithelial component, mutations in *TP53*, *PIK3CA*, *FBXW7*, *CHD4,* and *PPP2R1A* predominated [21]. Other studies of uterine carcinosarcomas containing a serous epithelial component also noted a similar mutational pattern [31,32].

The carcinoma component is probably the defining factor behind the tumor biology and aggressive nature, driving the trans differentiation into a sarcomatous component. Schiff et al. employed comparative genomic hybridization and fluorescence in situ hybridization (FISH) on 30 uterine and ovarian carcinosarcoma samples. They found significant gene amplification, particularly of *c-myc* within the carcinoma component. They also found a higher proliferation index in the carcinoma component using Ki67 immunohistochemistry. These findings suggest high chromosomal instability and support the more aggressive nature of the carcinoma component relative to the sarcoma component [33]. Moreover, in a recent study, Cuevas et al. were able to induce uterine carcinosarcomas from well-differentiated endometrioid carcinomas in a mouse model through the inactivation of FBXW7 and PTEN in epithelial cells. The genomic analysis of the tumors revealed that most tumors spontaneously acquired a *TP53* mutation, suggesting a potential synergistic role of the FBXW7, PTEN/PI3K, and p53 pathways in uterine carcinosarcoma tumorigenesis [34]. Their success in inducing and maintaining a uterine carcinosarcoma by manipulating the epithelial cells further supports the epithelial-driven theory of carcinosarcoma genesis. Two other studies lend further support to the idea that the carcinomatous component is the driver of tumor aggressiveness. A study by Emoto et al. investigated the differences in angiogenesis in the two parts of carcinosarcomas and found a higher VEGF and microvessel density in the epithelial component [35]. Another study likewise reported a higher apoptotic index in the sarcomatous than the carcinomatous component, supporting again that the carcinomatous element plays a key role in the aggressive behavior of this tumor [36]. Clinically, we see evidence of these findings in the fact that most metastatic lesions are carcinomatous in nature [8,14].

**Table 1 cancers-14-04465-t001:** Mutational profile summary for uterine carcinosarcomas [17,19,20,21,22,23,37,38].

Gene	Frequency
*TP53*	62–91%
*MLL3*	29%
*CSMD3*	23%
*H2A/H2B*	21%
*FBXW7*	19–39%
*PTEN*	18–41%
*BAZ1A, RPL22*	18%
*PIK3CA*	17–41%
*CTCF*	17%
*CCNE1*	16–40%
*FOXA2*	15%
*KMT2C*	13%
*ACVR2A*	12%
*PIK3R1*	11–23%
*CHD4*	11–17%
*ZBTB7B, JAK1,RAD50*	11%
*PPP2R1A*	10–28%
*ARID1A*	10–27%
*ATM, BCORL1*	10%
*KRAS*	9–27%
*RB1*	9–11%
*CREBBP, RNF43*	9%
*MSH2,PAPL, ABCC9, NF1, SPEN, INPPL1*	8%
*AKT3, CTCF, ERBB3, TNK2*	7%
*ZFHX3*	7–10%
*MSH6*	6–18%
*MLH1,C2CD2, BLM, MGA, CASP8, RASA1*	6%
*ATRX, LIMCH1, KMT2A*	5%
*ARHGAP35*	4–11%
*TAF1*	4–8%
*U2AF1, INSR, STAG2, KLF5, PLXNC1, RPS6KA3, BRIP1, RAD51C, AGO2, MBD4, TGFBR2*	4%
*SPOP*	3–18%
*CTNNB1*	3–12%
*EP300, FGFR2, MAP3K4, MED12, CCND1, AKT1, PIK3R2, GNAQ, B2M*	3%
*BRCA2*	2–15%
*BRCA1*	0–6%

In 2013, the Cancer Genome Atlas (TCGA) characterized endometrial carcinomas into four distinct molecular subgroups: *POLE* ultramutated, microsatellite instability hypermutated, copy number low, and copy number high [39]. Although uterine carcinosarcomas were not included in this analysis, Gotoh et al. subsequently attempted to classify uterine carcinosarcomas using these same molecular characteristics. While many separated into the copy-number-high (serous-like) subgroup, 47% were better classified in one of the three other groups [17]. Similarly, Travaligno et al. confirmed the applicability of the four subgroups in uterine carcinosarcoma, with the copy-number-high group being the predominant subset (91% of those included). Specifically, they found that subgroups with high mutational load (*POLE*-mutated types and those with high microsatellite instability (MSI-high)) were less common in uterine carcinosarcomas than in other endometrial cancers [25]. The prognostic value of the TCGA classification as applied to uterine carcinosarcoma showed that *POLE*-mutated carcinosarcomas have an excellent prognosis, similar to that of non-carcinosarcoma endometrial cancers. In contrast, though, the *TP53*-mutated (copy-number-high/serous-like group) and no specific molecular profile (surrogate of the copy-number-low/endometrioid-like group) groups were associated with a poorer prognosis than their endometrial carcinoma counterparts [40].

Uterine carcinosarcoma incidence rates are five times higher than those of ovarian carcinosarcomas, resulting in fewer studies focusing on ovarian carcinosarcomas in the literature [41,42]. Whereas uterine carcinosarcomas are heterogeneous in their molecular profiling, ovarian carcinosarcomas represent a more homogenous group [7]. In an analysis of the transcriptome of ovarian carcinosarcomas, Gotoh et al. found that the gene expression of ovarian carcinosarcomas most resembled that of high-grade serous ovarian cancer [17]. This is consistent with the fact that the common histology for the carcinoma component is high-grade serous histology [7]. Some studies have shown demonstrated the presence of homologous recombination deficiency in ovarian carcinosarcomas [17,43,44]. Taken together, these findings propose a distinct biological profile for uterine and ovarian carcinosarcomas despite their similar histologic appearance.

### 2.2. Differences from Pure Carcinomas

Despite the molecular overlap with carcinomas, the mutational profile of carcinosarcomas maintains some key differences that distinguish it from other tumors of the same anatomic site [23]. Unlike most endometrial tumors, the majority of uterine carcinosarcomas contain *TP53* and *PTEN* mutations simultaneously [20,39]. Carcinosarcomas also have been found to have a significantly higher whole-genome doubling than other tumor subtypes, with doubling occurring in 90% of the tumors [20,45]. This percentage is significantly higher than in uterine corpus endometrial carcinomas and ovarian serous carcinomas, which have 22% and 56% whole-genome doubling frequency, respectively [46]. What has really differentiated gynecologic carcinosarcomas from their carcinoma counterparts, however, has been their high rates of mutations in both chromatin-remodeling genes as well as in epithelial–mesenchymal transition (EMT) genes. Jones et al. performed whole-exome sequencing on 22 uterine and ovarian carcinosarcomas and revealed that carcinosarcomas demonstrate one of the highest rates of chromatin remodeling dysregulation of all tumors to date, occurring in approximately two-thirds of cases [22]. *ARID1A* and *ARID1B*, key players in the SWI/SNF chromatin remodeling complex, were frequently mutated. Other alterations were also reported, including mutations in histone methyltransferase *MLL3*, tumor suppressor *SPOP*, and *BAZ1A*, a component of the chromatin assembly factor [22]. Mutations in these epigenetic regulators have been shown to be associated with important clinical outcomes in different tumors [47,48]. For instance, mutations in *ARID1A* and *ARID1B* have been associated with a decreased survival in patients with neuroblastoma [48]. Their significance in gynecologic carcinosarcomas remains to be elucidated.

Building on this theme, Zhao et al.’s whole-exome sequencing of 68 uterine and ovarian carcinosarcomas confirmed the high mutation rates in histone genes, particularly in genes coding for histone H2A and H2B. They also noted an amplification of the segment of chromosome 6p that contains the histone gene cluster of these genes. When carcinosarcoma cell lines were transfected with mutant H2A and H2B genes, EMT markers showed an accompanying increase, and there was an upregulation in tumor migration and invasion. These findings suggest a potential regulatory role of histones and chromatin remodelers in EMT and sarcomatous transdifferentiation [21].

EMT is a reversible process that involves the transition of a cell from an epithelial to a mesenchymal phenotype. In the context of cancer, EMT plays a critical role in tumor initiation, progression, metastasis, and invasion [49]. This is regulated by a complex system that includes transcriptional factors such as Snail, SLUG, and ZEB1 and ZEB2 and multiple other players such as TGF-B, the JAK/STAT pathway, and microRNAs [50,51]. Particularly, the downregulation of miR-200 family members has been implicated in various aggressive tumors, which is secondary to their role as strong inhibitors of EMT, tumor invasion, and metastasis, among others [51]. The role of EMT in carcinosarcoma has been well-studied, with a special focus on its role in the transdifferentiation of the carcinomatous component into the sarcomatous component. Carcinosarcomas are now widely regarded as one of the best examples of stable EMT [52,53]. Multiple studies have confirmed an association between uterine carcinosarcoma and EMT and show an upregulation in EMT-related genes compared with endometrial carcinomas [20,54,55].

Studies have also confirmed a higher EMT score in the sarcomatous component compared with the epithelial part using both IHC and RT-PCR [32,56,57]. Using transcriptome sequencing, Cherniack et al. found a positive correlation between the EMT score and the presence of heterologous sarcoma histologies as well as a correlation with an increasing proportion of the tumor being made up of a sarcoma component [20]. This study also explored the potential regulators of EMT in uterine carcinosarcoma and reported a key role for the miR-200 family, where its downregulation via promoter hypermethylation is correlated to higher EMT scores and to the presence of a sarcomatous element [20]. Gotoh et al. studied the transcriptome and DNA methylome of carcinosarcomas to confirm the association between EMT and the development of a sarcomatous component. They did not find any correlation between EMT scores and the molecular subtypes of uterine carcinosarcomas previously described [17]. Interestingly, they also did not find an association between EMT and CTNNB1-activating mutations, which have been associated with an induction of EMT in various other tumors [58,59]. Rather, they found *CTNNB1* to be associated with the hypomethylation of members of the miR200 family, a finding that was somewhat paradoxical [17]. This likely reflects the complexity of EMT regulators on both a genetic and epigenetic level.

Several other genes have been implicated in EMT in carcinosarcomas. These include *HMGA2* as a regulator of Snail expression and the expression of its downstream effectors and *ALK* as an inducer of EMT and inhibitor of apoptosis [56,60]. It is worth noting that the majority of the molecular and genetic studies investigating EMT in carcinosarcomas have been mostly limited to uterine carcinosarcomas, and very few include ovarian carcinosarcomas. Whether these molecular findings apply to ovarian carcinosarcomas remains to be seen. Lastly, one important corollary to the above discussion is a better understanding of the differences between the carcinomatous and sarcomatous components within the same tumor. Some differences may be due to EMT, but there are likely some other differences between the carcinomatous and sarcomatous components that are present that are independent of EMT. This molecular information may ultimately provide important insights into novel approaches for treatments of this highly aggressive but somewhat unique endometrial carcinoma.

## 3. Immune Microenvironment in Carcinosarcoma

Several studies have elucidated the integral role of the tumor immune microenvironment in influencing tumor behavior—from genesis to invasion to metastasis—as well as in regard to response to treatment across numerous tumor types [61,62]. Although studies on gynecologic carcinosarcomas are less abundant, there are a variety of studies that have evaluated the immune microenvironment in uterine and ovarian carcinomas. Building on the above work in molecular profiling, studies in endometrial cancer have explored the association between TCGA classifications and the immune microenvironment. Specifically, *POLE*-mutated and MSI-high tumors are associated with a higher tumor mutational burden, likely secondary to the large number of neoantigens resulting from the hypermutated state [63]. In turn, this has been associated with a more robust antitumor response and higher infiltrates of T and B cells, a correlation shown in other tumor types as well [64,65]. In contrast to some subgroups of endometrial cancer, however, epithelial ovarian carcinomas have a low tumor mutational burden [66]. Several studies have demonstrated the prognostic implications of tumor mutational burden and immune cell infiltration in ovarian tumors [67,68]. Both higher tumor mutational burden and increased levels of immune cell infiltration suggest a more immunologically hot microenvironment, and patients with these tumors tend to have improved prognoses. However, the immune microenvironment of most epithelial ovarian carcinomas is relatively immunosuppressed, and various studies have explored the mechanisms for its relative immune resistance [69]. Whether this is similarly true in ovarian carcinosarcomas has not yet been established.

Likely due to its relative rarity, however, fewer studies have provided an in-depth analysis of the immune microenvironment of gynecologic carcinosarcomas. In arguably the most comprehensive evaluation of the immune microenvironment, Gotoh et al. investigated the role of the immune cells infiltrating 100 gynecologic carcinosarcoma specimens in relation to their previously published molecular subtypes discussed above [17,20]. As expected, they first demonstrated that MSI-high tumors had a higher infiltration of immune cells, particularly CD8+ T cells, activated memory CD4+ T cells, M1 (pro-inflammatory) macrophages, and plasma cells when compared with copy-number-high tumors [19]. A subsequent evaluation of transcriptomics on the immune microenvironment on these samples using 2650 immune-related genes was used to subclassify gynecologic carcinosarcomas into four immune microenvironmental subtypes (ISs). IS1 was the “immunologically hot” subtype, with high infiltration of activated immune cells. This subtype also had a disproportionately higher number of hypermutated tumors (*POLE*-mutated and MSI-high subsets). This subtype was also the one with the highest diversity of T-cell receptors, an attribute previously associated with improved survival outcomes [70,71]. IS2 had fibroblasts, endothelial cells, and M2 (anti-inflammatory) macrophages. IS3 consisted mainly of activated NK cells and memory B cells, and IS4 had high infiltrates of M0 macrophages and naïve T cells. Both IS2 and IS4 had a higher proportion of copy-number-high tumors that were not hypermutated [19]. While these findings suggest a potentially useful immune type subclassification and may be useful for predicting immune therapy responsiveness, more studies are awaited to validate these findings and ensure reproducibility.

Other smaller studies have also investigated questions concerning the immune microenvironment. Karpathiou et al. explored the expression of various immune-related genes in the primary and metastatic carcinosarcomas of gynecologic origin using immunohistochemistry [72]. This study showed a higher proportion of CD3 expression in the sarcomatous component compared with the carcinoma. CTLA-4, an immune checkpoint that downregulates T cells, was noted to be higher in the carcinoma counterpart. Although this group found no PD-L1 expression in any of the samples, previous studies have reported a 25% expression in uterine carcinosarcomas and up to 50% in ovarian carcinosarcomas [18,73]. One such study also reported that PD-L1 expression in the sarcomatous component was associated with shorter overall survival, as well as a decrease in CD8+ lymphocytic infiltration in the sarcomatous but not the carcinomatous component [73].

These differences in immune cell infiltrates are further supported by the data from de Silva et al., who reported that there was a greater degree of infiltration of most of the immune markers in the sarcomatous portion compared with the epithelial portion (CD3, CD4, CD8, FOXP3, PD-1, and PD-L2) [74]. However, in direct contrast to the data from Zhu et al., they reported improved overall survival in patients whose tumors demonstrated higher PD-L1 expression in the sarcomatous component. They also reported a more favorable OS in those patients whose tumors had high carcinomatous PD-1 and PD-L1 expression [74]. Related work in pulmonary carcinosarcomas evaluated tumor-infiltrating lymphocytes [75]. These studies showed higher T-cell infiltrates, including CD3+ cells and macrophages as well as a higher PD-L1 expression relative to other non-small cell lung carcinomas in this tumor subtype [76,77].

Varying thresholds for defining marker positivity, differing proportions of epithelial and sarcomatous histologies, and small sample sizes might contribute to the inconsistent findings. This could be reflective of the larger molecular complexity of carcinosarcomas. First, there are multiple histologic subtypes that can make up both the epithelial and the sarcomatous components. Second, these components can, however, be represented in varying proportions. Multiple levels of heterogeneity make it difficult to characterize these tumors as one cohesive group, and thus data are likely to be similarly heterogeneous depending on the specific tumors represented in the cohort. Still, enough molecular similarities exist that it is likely that some immune-based themes will also emerge for gynecologic carcinosarcomas.

The authors also looked at the immunological profiles in the carcinoma versus the sarcoma component using RNA-seq and found distinct immune cell populations present in each of the two components. The sarcoma component was found to have a higher infiltration of T cells, T-cell-mediated tumor cell death, as well as plasma cells, M2 macrophages, and fibroblasts compared with the carcinoma component. This simultaneous increase in immune-suppressive M2 macrophages and immune-stimulatory T cells and plasma cells is intriguing and warrants further investigation. In the carcinoma component, they found a higher abundance of M0 (non-activated) macrophages and follicular helper T cells. Interestingly, the sarcoma component seems to have a higher T-cell receptor diversity as well [19]. Whether this has implications for the likelihood of response to immunotherapy is yet to be determined.

A large study by Luke at al. explored the association between the Wnt/B-catenin pathway activation and the tumor immune microenvironment across 31 solid human tumors. Interestingly, uterine carcinosarcomas had among the highest non-T-cell-inflamed phenotype of the cohort [78]. This phenotype has been shown to be negatively correlated with response to immunotherapy [79,80]. It is worth noting, however, that in their study, uterine carcinosarcomas were among the few non-T-cell-inflamed tumors without an associated Wnt/B-catenin pathway activation [78]. This suggests that there may be different driver pathways in this tumor subtype that result in the immune exclusion phenotype.

Another relevant protein is the CKLF-like MARVEL transmembrane domain-containing 6 (CMTM6), which is known to be a regulator and stabilizer of PD-L1 expression [81,82]. Its role in the tumor immune microenvironment is the subject of much intrigue in various tumors, primarily regarding the response to PD-1/PD-L1 inhibitors [83,84]. In a large study by Zhao et al., CMTM6 expression was analyzed across 33 different tumor types. Their cohort included 56 uterine carcinosarcomas extracted from the TCGA cohort, and the group reported a significant correlation between CMTM6 expression and PD-L1 protein expression in uterine carcinosarcoma. Uterine cancers contained the highest proportion of CMTM6 mutations among all the tumors studied [85].

Huang et al. demonstrated that primary tumors that had not metastasized had relatively more mast cells than metastatic tumors [86]. They analyzed metastasis-related genes such as adenylate kinase-8 (AK8) and myelin protein zero (MPZ) and found that both AK8 and MPZ were associated with activated mast cells in these samples. The suppression of AK8 and MPZ via siRNA cell transfection on uterine carcinosarcoma cell lines led to the enhanced migration and invasion of carcinosarcoma cells in trans-well and wound healing assays. Based on their findings, they proposed a mechanistic role for AK8, MPZ, and activated mast cells in the invasion and metastasis of carcinosarcomas. However, the role of mast cells as either a protective antitumor factor or a driving factor for tumorigenesis and progression remains unsettled and might have differential effects contingent on the tumor site [87,88]. This requires further studies to confirm and expand on the role of mast cells, if any, in gynecologic carcinosarcomas.

## 4. Clinical Data for Immunotherapy

Although immunotherapy has revolutionized the treatment of multiple cancer types, particularly in the advanced setting, less is known about its role in gynecologic carcinosarcomas [89]. With an increasing understanding of the molecular and immunologic phenotypes of carcinosarcomas, we believe that immunotherapy may have a role and may be part of the key to improving outcomes in this aggressive tumor subtype.

Immune checkpoint inhibitors are at the forefront of immunotherapy, including PD-1, PD-L1, and CTLA-4 inhibitors. Single-agent immune checkpoint inhibitors have had success in MSI-high endometrial carcinomas [90,91], and now both pembrolizumab and dostarlimab have relevant indications. Pembrolizumab is also approved for those patients whose tumors demonstrate a high tumor mutational burden, such as those with *POLE* mutations [92]. The two case reports of patients with recurrent uterine carcinosarcomas demonstrating these biomarkers who were successfully treated with pembrolizumab are encouraging [93,94].

Unfortunately, single-agent checkpoint inhibitors had limited success in endometrial and ovarian tumors without these biomarkers of high tumor mutational burden [95,96]. In general, PD-L1 positivity has not been a promising predictive biomarker for ovarian and endometrial cancers [97,98], although a case of a patient with a PD-L1 positive pulmonary carcinosarcoma successfully treated with nivolumab has been reported [99]. In patients with microsatellite stable endometrial tumors, the combination of pembrolizumab and lenvatinib, a tyrosine kinase inhibitor, has been more successful [100,101]. This combination is now FDA-approved for patients with advanced endometrial carcinoma that is not MSI-high or MMR-deficient and does not specify histologic subtypes. This approval is based upon the data from Makker et al., who reported a 36% response rate in patients with microsatellite stable endometrial cancers to this combination [100,101]. While the exact number of carcinosarcoma patients included in this trial is unknown, these patients were not excluded. Furthermore, the response rates in patients with other high-grade endometrial carcinomas were similar to those in patients with endometrioid tumors, which further supports the use of this combination [100]. Subsequently, How et al. published their single-institution experience with pembrolizumab and lenvatinib and reported an objective response rate of 25% and a clinical benefit rate of 58% in patients with uterine carcinosarcomas [102]. Qualitatively, this is a notably high response rate for second and subsequent-line treatments in recurrent carcinosarcomas given the low response rates seen in the recurrent setting in general.

Patients with gynecologic carcinosarcomas were enrolled in some of the other endometrial cancer studies that have been completed. The Z1D subprotocol of the NCI-MATCH trial evaluated the efficacy of nivolumab, another PD-1 inhibitor, in MMR-deficient non-colorectal cancers. The subprotocol cohort consisted of 42 eligible patients with 4 uterine carcinosarcoma patients. The overall objective response rate (ORR) was 36% (90% CI, 23.5% to 49.5%), and the median overall survival was 17.3 months. In the carcinosarcoma cohort, two out of four patients achieved some clinical benefit: One patient had a partial response, and one patient had stable disease [103]. KEYNOTE-028 was a multicohort phase Ib study aimed to investigate pembrolizumab’s safety and efficacy in PDL1-positive advanced solid tumors and evaluated 24 patients with endometrial cancer. Although one patient with carcinosarcoma was enrolled, there were no specific outcome data available for this patient, and the objective response rate in this study was low overall [97].

There are also several ongoing phase I, II, and III clinical trials evaluating immune checkpoint inhibitors, either alone or in combination, for patients with gynecologic malignancies that allow for the enrollment of patients with carcinosarcomas. Representative trials are listed in Table 2. We hope that by allowing for the registration of patients with gynecologic carcinosarcomas into trials evaluating other epithelial tumors of the same tumor site, we can increase our knowledge of the role of immunotherapy in this patient population and gain further insight into tumor biology.

## 5. Future Directions

As with other tumor types, finding ways to improve the efficacy of immunotherapy will be valuable to the development of new treatments. Less work has been carried out regarding the mechanisms for immunotherapy response and resistance in gynecologic carcinosarcomas relative to many other cancer types, such as lung cancer, breast cancer, and colorectal cancer. EMT has been found to have implications for the immune response in other tumor types [104,105]. Mechanistically, the cause and effect of the interaction between EMT and immune cells are complex. Both innate and adaptive immune cells have been implicated in the development of EMT, though their roles in the development and progression of EMT are varied and are largely context-dependent [106,107]. Research from other tumor types suggests that EMT progression is mediated by immune regulation; specifically, cytokines such as TGF-beta, IL-10, and immune cell populations such as tumor-associated macrophages and some subsets of CD4+ and CD8+ T cells appear to be critical to this regulation [108,109,110,111,112]. Although the upregulation of the Wnt/B-catenin pathway has been associated with both EMT [59] and immune exclusion [78,113,114] in other tumors, its importance in carcinosarcoma is less clear [78]. Given the prominent role of EMT in gynecologic carcinosarcomas, however, it is likely that understanding the role that EMT plays in the interactions between tumor cells and the immune response will be important to identify future treatment approaches.

Ovarian and uterine carcinosarcomas are, by definition, carcinomas of the ovary and endometrium, respectively. Therefore, it is likely that many of the same barriers to immunotherapy that are present for other ovarian carcinomas and microsatellite stable/low tumor mutational burden endometrial carcinomas may be present for gynecologic carcinosarcomas. For example, future research addressing the mechanisms for the lower tumor mutational burden found in epithelial ovarian tumors [115] and the higher rates of PTEN loss in endometrial cancer [116], may ultimately be beneficial for gynecologic patients carcinosarcomas as well. Given the relative rarity of carcinosarcomas, we hope that future trials of novel immunotherapy agents in patients with ovarian and endometrial carcinomas will not exclude patients with carcinosarcomas from enrollment.

## 6. Conclusions

Ovarian and uterine carcinosarcomas encompass a heterogeneous set of tumors, and patients with this rarer subtype have worse survival outcomes than those with other epithelial tumors of the same origin. By improving our understanding of the molecular and immunologic landscapes of gynecologic carcinosarcomas, we hope that future treatments will be developed that take advantage of the unique features of these tumors. Further research is needed to understand the complex immune microenvironment in the many molecularly distinct subtypes of these tumors and find the overarching themes and treatment approaches that target those features specific to the evolution of carcinosarcomas from epithelial tumors.

## Figures and Tables

**Figure 1 cancers-14-04465-f001:**
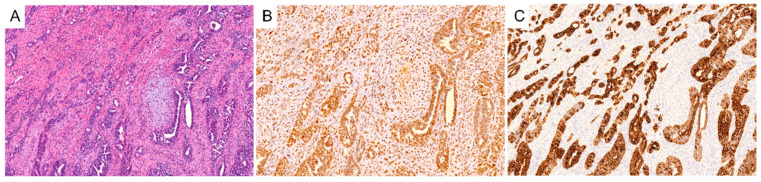
Representative hematoxylin and eosin section of a uterine carcinosarcoma (**A**) and immunohistochemical expression of P53 (**B**) and cytokeratin-7 (**C**) with a serous epithelial component.

**Table 2 cancers-14-04465-t002:** Selection of ongoing immunotherapy clinical trials open to accrual of patients with gynecologic carcinosarcomas (as of July 2022).

Trial	Phase	Target Condition	Treatment Arms	Immunotherapy Mechanism of Action
Uterine Carcinosarcomas				
NCT04906382	I	Recurrent MMRd Endometrial Cancer	Tislelizumab	Tislelizumab: anti-PD-1 mAb
CA017-056 NCT04106414	II	Recurrent/Persistent Endometrial Carcinoma or Endometrial Carcinosarcoma	Arm 1: Nivolumab + BMS-986205 (IDO-inhibitor) Arm 2: Nivolumab alone	Nivolumab: anti-PD-1 mAb
NCT05156268	II	Recurrent/Persistent Endometrial Carcinoma or Endometrial Carcinosarcoma	Pembrolizumab + Olaparib	Pembrolizumab: anti-PD-1 mAb
EndoBARR NCT03694262	II	Recurrent Endometrial Cancer	Atezolizumab + Bevacizumab + Rucaparib	Atezolizumab: anti-PD-L1 mAb
NCT03241745	II	MSI/MMRd/Hypermutated Uterine Cancer	Nivolumab	Nivolumab: anti-PD-1 mAb
NCT05147558	II	Advanced Uterine Carcinosarcoma	Pembrolizumab +Lenvatinib	Pembrolizumab: anti-PD-1 mAb
NCT03015129	II	Recurrent/Persistent Endometrial Carcinoma	Arm 1: Durvalumab + Tremelimumab Arm 2: Durvalumab	Durvalumab: anti-PD-L1 mAb Tremelimumab: anti-CTLA4 mAb
ACROPOLI NCT04802876	II	PD1-high-expressing Tumors	Spartalizumab	Spartalizumab: anti-PD-1 mAb
DART NCT02834013	II	Advanced Rare Tumors	Arm 1: Nivolumab + Ipilimumab Arm 2 (PD-L1 amplified cohort): Nivolumab	Nivolumab: anti-PD-1 mAb Ipilimumab: anti-CTLA4 mAb
AtTEnd NCT03603184	III	Advanced/Recurrent Endometrial Cancer	Arm 1: Atezolizumab + Paclitaxel + Carboplatin Arm 2: Paclitaxel + Carboplatin	Atezolizumab: anti-PD-L1 mAb
RUBY NCT03981796	III	Advanced/Recurrent Endometrial Cancer	Part 1: Arm 1: Dostarlimab + Paclitaxel + Carboplatin Arm 2: Paclitaxel + Carboplatin Part 2: Arm 1: Dostarlimab + Paclitaxel + Carboplatin + Niraparib Arm 2: Paclitaxel/Carboplatin	Dostarlimab: anti-PD-1 mAb
GOG-3053 NCT04634877	III	Newly Diagnosed High-Risk Endometrial Cancer	Arm 1: Pembrolizumab + Paclitaxel/Carboplatin Arm 2: Paclitaxel/Carboplatin	Pembrolizumab: anti-PD-1 mAb
Ovarian Carcinosarcomas				
NCT04919629	II	Recurrent Ovarian, Fallopian Tube or Primary Peritoneal Cancer and Malignant Effusion	Arm 1: APL-2 (Pegcetacoplan) and pembrolizumab Arm 2: APL-2 and pembrolizumab Arm 3: Bevacizumab only	Pembrolizumab: anti-PD-1 mAb
BRIGHT NCT05044871	II	Recurrent Platinum-resistant Epithelial Ovarian Cancer	Arms relevant to patients with carcinosarcoma: Arm 1: (≥3 CD8+ TILs mucinous ovarian cancer and ovarian carcinosarcoma cohort): Tislelizumab + Bevacizumab + Nab-paclitaxel Arm 2: (<3 CD8+ TILs mucinous ovarian cancer and ovarian carcinosarcoma cohort): Bevacizumab + Nab-paclitaxel	Tislelizumab: anti-PD-1 mAb
GOG-3036 NCT03740165	III	Advanced Epithelial Ovarian Cancer	Arm 1: Pembrolizumab + Olaparib + Paclitaxel + Carboplatin Arm 2: Pembrolizumab + Paclitaxel + Carboplatin Arm 3: Paclitaxel + Carboplatin	Pembrolizumab: anti-PD-1 mAb
DUO-O NCT03737643	III	Newly Diagnosed Advanced Ovarian Cancer	Arm 1: Durvalumab + Bevacizumab + Olaparib + Paclitaxel + Carboplatin Arm 2: Durvalumab + Bevacizumab + Paclitaxel + Carboplatin Arm 3: Bevacizumab + Paclitaxel + Carboplatin	Durvalumab: anti-PD-L1 mAb
Uterine and Ovarian Carcinosarcomas				
NCT05224999	II	Recurrent/Metastatic Carcinosarcomas	Nivolumab	Nivolumab: anti-PD-1 mAb
NCT05265793	II	Advanced Sarcomatoid Carcinoma or Carcinosarcoma	Camrelizumab + Apatinib	Camrelizumab: anti-PD-1 mAb
ROCSAN NCT03651206	II/III	Recurrent/Metastatic Ovarian and Endometrial Carcinosarcomas	Arm 1: Dostarlimab + Niraparib Arm 2: Niraparib Arm 3: SOC Chemotherapy	Dostarlimab: anti-PD-1 mAb
MOST-CIRCUIT NCT04969887	II	Immunotherapy Sensitive Advanced Rare Cancers	Nivolumab + Ipilimumab	Nivolumab: anti-PD-1 mAb Ipilimumab: anti-CTLA4 mAb

MSI, microsatellite instability; MMRd, mismatch repair deficient; TCR, T-cell receptor; TIL, tumor-infiltrating lymphocytes; mAb, monoclonal antibody; PD-1, programmed cell death protein 1; PD-L1, programmed cell death ligand 1; CTLA-4, cytotoxic T-lymphocyte-associated protein 4.

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
