# Peer review of "Molecular Characterizations of Gynecologic Carcinosarcomas: A Focus on the Immune Microenvironment"

_cancers, 2022, doi:10.3390/cancers14184465_

Round 1
Reviewer 1 Report
This review is high impact and relates to cutting-edge topic, molecular subtyping and immunotherapy in carcinosarcomas. Due to shifting paradigm of carcinasarcoma pathogenesis and current view which describe caricnosarcoma as an epithelial tumor instead of mixed one, the therapeutic algorithms also should be changed. The immune microenvironment plays a crucial role in platinum-based and immunotherapy sensitivity so it is dramatically important to be aware about carcinosarcoma immune cells peculiar features. This review covers all points of carcinosarcoma including classification, molecular subtyping application, immune microenvironment, immunotherapy strategies and future directions. The references are a little bit excessive so the authors could focus on the most impactful ones. At the same time the authors listed as fundamental as the most recent papers.
As for more detailed notes I should mention that table one does not demonstrate clearly the localization and components of carcinosarcomas although in the text the authors mentioned that these tumors can differ a lot due to different localization and histological characteristics. I recommend to include this information into table 1 to give it as a guide for the readers. In addition, I recommend to point the mechanisms of the treatment agents in table 2. For example, Tislelizumab is a monoclonal antibody directed against PD-1.
After this minor correction this article could be recommended for publication.
Author Response
Thank you for your comments. Please find below the responses:
- Comment 1: The references are a little bit excessive so the authors could focus on the most impactful ones. At the same time the authors listed as fundamental as the most recent papers.
Response: We thank the reviewer for this comment. We have reduced the number of references where redundancy existed, while keeping the most impactful studies. - Comment 2: As for more detailed notes I should mention that table one does not demonstrate clearly the localization and components of carcinosarcomas although in the text the authors mentioned that these tumors can differ a lot due to different localization and histological characteristics. I recommend to include this information into table 1 to give it as a guide for the readers.
Response: This is an important point and we thank the reviewer for this insightful comment. Unfortunately, to our knowledge, these data about differences in mutational profiles of carcinosarcomas with different epithelial subtypes are not currently available. We too think this is an interesting question. However, conceptually, we believe that at least in part this finding explains why different studies report varying distribution of mutations as shown in table 1. Carcinosarcomas are a very heterogeneous tumor type that has been grouped together historically, but it is possible that paradigms will change in the future. - Comment 3: In addition, I recommend to point the mechanisms of the treatment agents in table 2. For example, Tislelizumab is a monoclonal antibody directed against PD-1.
Response: This important point has been incorporated into Table 2. We also removed the expected accrual to streamline this table further.
Reviewer 2 Report
Borrego et al have give a comprehensive review on gynecologic carcinosarcoma with a special focus on immune microenvironment. Overall, the review is well written with adequate details on immune microenvironment section. Although the article is focused on immune microenvironment, the authors could further elaborate Section 2. Molecular characterization in detail (in particular Section 2.2) to enrich the quality of the review. In that condition I recommend the article to considered for publication.
Author Response
Thank you for your comments. Please find below the responses:
- Comment: Although the article is focused on immune microenvironment, the authors could further elaborate Section 2. Molecular characterization in detail (in particular Section 2.2) to enrich the quality of the review. In that condition I recommend the article to considered for publication.
Response: Additional details have been added to Section 2.2. We also changed the title of this section to better reflect the content and message being conveyed.